# Multi-Site MRI Data Harmonization with an Adversarial Learning Approach: Implementation to the Study of Brain Connectivity in Autism Spectrum Disorders

**Federico Campo [1,2], Alessandra Retico [2,*], Sara Calderoni [3,4] and Piernicola Oliva [5,6]**

[1] Department of Physics, University of Pisa, 56127 Pisa, Italy; f.campo7@studenti.unipi.it
[2] National Institute for Nuclear Physics (INFN), Pisa Division, 56127 Pisa, Italy
[3] Developmental Psychiatry Unit, IRCCS Stella Maris Foundation, 56127 Pisa, Italy; sara.calderoni@fsm.unipi.it
[4] Department of Clinical and Experimental Medicine, University of Pisa, 56127 Pisa, Italy
[5] Department of Chemical, Physical, Mathematical and Natural Sciences, University of Sassari, 07100 Sassari, Italy; oliva@uniss.it
[6] INFN, Cagliari Division, 09042 Monserrato (Cagliari), Italy
*    Correspondence: alessandra.retico@pi.infn.it

**Abstract:** Magnetic resonance imaging (MRI) nowadays plays an important role in the identification of brain underpinnings in a wide range of neuropsychiatric disorders, including Autism Spectrum Disorders (ASD). Characterizing the hallmarks in these pathologies is not a straightforward task and machine learning (ML) is certainly one of the most promising tools for addressing complex and non-linear problems. ML algorithms and, in particular, deep neural networks (DNNs), need large datasets in order to be properly trained and thus ensure generalization capabilities on new data. Large datasets can be obtained by collecting images from different centers, thus bringing unavoidable biases in the analysis due to differences in hardware and scanning protocols between different centers. In this work, we dealt with the issue of multicenter MRI data harmonization by comparing two different approaches: the analytical ComBat-GAM procedure, whose effectiveness is already documented in the literature, and an originally developed site-adversarial deep neural network (ad-DNN). The latter aims to perform a classification task while simultaneously searching for site-relevant patterns in order to make predictions free from site-related biases. As a case study, we implemented DNN and ad-DNN classifiers to distinguish subjects with ASD with respect to typical developing controls based on functional connectivity measures derived from data of the multicenter ABIDE collection. The classification performance of the proposed ad-DNN, measured in terms of the area under the ROC curve (AUC), achieved the value of AUC = $0.70 \pm 0.03$, which is comparable to that obtained by a DNN on data harmonized according to the analytical procedure (AUC = $0.71 \pm 0.01$). The relevant functional connectivity alterations identified by both procedures showed an agreement between each other and with the patterns of neuroanatomical alterations previously detected in the same cohort of subjects.

**Keywords:** brain connectivity; machine learning; adversarial learning; autism spectrum disorders (ASD); multi-site harmonization; explainable AI (XAI)

## 1. Introduction

Functional MRI (fMRI) is a valid instrument in studies on neurodevelopmental disorders since it allows for the investigation of physiological changes arising from blood flow variations across the brain structure, which are strictly linked with neuronal activity. fMRI acquired during the resting state of an individual is referred to as resting-state fMRI (rs-fMRI), and allows for the study of altered functional connectivity networks in subjects with neurodevelopmental disorders, including autism spectrum disorders (ASD) (for a recent meta-analysis, see Lau et al. [1]).

ASD is a diagnostic category of neurodevelopmental disorders characterized by impaired social interaction and communication, as well as the presence of restricted interests, repetitive behavior and alterations in sensory processing [2]. So far, no univocal biomarkers have been identified for this disorder, on the basis of a recent systematic review encompassing biochemical, genetic, neuroimaging, neurophysiological and neuropsychological measures [3]. As of now, a big percentage of neuroimaging studies have been carried out with limited data acquired from a single medical center; this approach, while minimizing variability in data resulting from the different instrumentation employed, also reduces statistical power for detecting the effects of disease and their moderators [4]. To overcome this hurdle, in recent years, the common approach is to create larger datasets by pooling together data acquired from different centers, including the Autism Brain Imaging Data Exchange (ABIDE) [5], the National Database for Autism Research (NDAR) (https://ndar.nih.gov, accessed on 5 April 2023) and the ENIGMA (Enhanced Neuroimaging Genetics Through Meta-Analysis) ASD Working Group [6] initiatives. However, this process has some downsides due to the introduction of variability between data. Indeed, data acquired from different sources can be affected by the so-called *batch effect*, which is related to differences in the acquisition system, such as the scanner model or acquisition parameters.

In order to mitigate the batch effect, a common harmonization procedure is the so-called ComBat (named after combating batch effects), and works by harmonizing data through location and scale adjustments. Its effectiveness was demonstrated on different datasets such as genes microarray data [7], diffusion tensor imaging [8] or structural [9] and functional MRI data [10].

The analysis of data belonging to different domains has also been addressed using techniques known as "domain adaptation", which is usually tested using data belonging to two different domains, referred to as *source* and *target* domain. Both domains consist of data of two different classes $A$ and $B$ and, while the entire *source* dataset is labeled, the *target* domain contains unlabeled data. The objective of the network is to learn distinctive features concerning classes from the *source* domain, as well as distinctive traits between the two domains, and to be able to generalize well from one domain to another. Ganin et al. [11] first designed this network and tested their algorithm using the *inter-twinning moons 2D* dataset as well as other popular image datasets, such as *The Office* dataset (http://ai.bu.edu/adaptation.html, accessed on 5 April 2023). Their network was able to learn distinctive patterns between classes from data belonging to the source dataset. At the same time, it was also trained to learn distinctive characteristics between data from the source and target domains. Combining the information about classes and domains, they taught the network to recognize important features for classification, as well as to generalize this information from one domain to another. The key of this network is a gradient reversal layer placed at the top of the domain classifier branch. It exploits a classic stochastic gradient descent procedure to update weights; however, during the training of this network, the forward propagation is not influenced by it, while, during back-propagation, it acts by multiplying the gradient by a negative parameter $-\lambda$ before passing it to the preceding layer. In this way, the partial derivatives of the site loss with respect to the parameters obtain a minus sign. The network is then encouraged to learn classes-related features and discouraged to learn the domain-related ones. They called this structure the domain-adversarial neural network (DANN). Kamath et al. [12] employed a network with a similar structure for natural language processing problems using different datasets, such as the Quora question pairs (https://www.kaggle.com/c/quora-question-pairs, accessed on 5 April 2023), the StackExchange dataset (https://archive.org/download/stackexchange, accessed on 5 April 2023) or the Stanford Natural Language Inference (https://nlp.stanford.edu/projects/snli, accessed on 5 April 2023). Guan et al. [13] obtained the adversarial evolution using a loss function resulting from the combination of two terms: one for the category classification and one for the domain classification. They employed their network for the analysis of brain MRI images from the ADNI (https://adni.loni.usc.edu, accessed on 5 April 2023) dataset.

As the case study in our work, we used the Autism Brain Imaging Data Exchange (ABIDE), a multi-site collection of data released for the first time in 2014 for the investigation of ASD. It contains structural and functional brain scans belonging either to individuals with ASD or to typically developing control subjects (TD). Given the multi-site nature of ABIDE, a harmonization procedure is usually implemented to minimize the batch effect [14].

In this study, we compared two different approaches to the classification of functional connectivity data of subjects with ASD and TD while mitigating biases related to the acquisition instrumentation (batch effect). As described in Section 2.3, the first step involves the computation of a brain connectivity measure known as Pearson correlation. Starting from these measures, we followed two different techniques. The first approach, as detailed in Section 2.4, involves the analytical harmonization technique known as ComBat, which we implemented by means of its state-of-the-art improved version, NeuroHarmonize (https://github.com/rpomponio/neuroHarmonize, accessed on 5 April 2023). The approach that we propose utilizes statistical corrections to remove site-related biases and involves the development of a deep neural network for the classification of harmonized data. The second approach, as detailed in Section 2.5, involves the development and use of a domain-adversarial neural network. This approach aims to discriminate between ASD and TD features while minimizing the biases caused by the sites through a learning process that minimizes the information learned from each site. In addition, we implemented an explainable AI algorithm called SHAP (https://github.com/slundberg/shap, accessed on 5 April 2023), as described in Section 2.6, to identify the brain areas that have more influence on the outcome of the deep learning classifier and quantify the importance of each feature. By identifying the altered functional connections related to these areas, we can determine the distinctive traits of subjects with ASD. Overall, our study aims to provide an insight into the performance and effectiveness of the two different approaches in mitigating site-related biases and classifying subjects with ASD with respect to control subjects.

## 2. Materials and Methods

### 2.1. The ABIDE Dataset

We used data collected within the Autism Brain Images Data Exchange (ABIDE) project, which was founded with the aim of investigating ASD by means of structural magnetic resonance images and resting-state fMRI (rs-fMRI) scans (https://fcon_1000 .projects.nitrc.org/indi/abide, accessed on 5 April 2023), and was published in two releases: ABIDE I, released in August 2012, which contains data of 1112 subjects (539 ASD and 573 TD) [5], and ABIDE II, released in June 2016, which contains data of 1114 subjects (593 ASD and 521 TD) [15]. ABIDE I includes scans collected at 17 different sites, whereas ABIDE II includes scans collected at 16 different sites.

For some sites, in both of these datasets, there are different releases of data from the same site, and, to denote them, they are usually marked with an underscore and the corresponding release number, such as "SITE_1" and "SITE_2". For the purposes of this work, we considered different batches of data from the same site as belonging to two different sites. This choice was made because different scanner or acquisition parameters could have been employed from an acquisition session to another within the same medical center. Following this procedure, we obtained a dataset comprising a total number of 38 sites. Table 1 displays the name of each site, the corresponding number of subjects (ASD and TD) provided by each site and the associated release (ABIDE I or II). The histograms showing the number of subjects with ASD and TD controls per site and the age distribution of the ABIDE dataset are shown in Figure 1.

**Table 1.** Number of subjects for each site of the ABIDE I and ABIDE II collections. The site IDs and the extended site names are reported, together with the sample composition in terms of the number of subjects with autism spectrum disorders (ASD) and the typically developing controls (TD), and the total number of subjects.

| Site ID | Site Name | ABIDE | Total (ASD, TD) |
|---------|-----------|-------|-----------------|
| CALTECH | California Institute of Technology | 1 | 38 (19, 19) |
| CMU | Carnegie Mellon University | 1 | 27 (14, 13) |
| KKI | Kennedy Krieger Institute | 1 | 55 (22, 33) |
| LEUVEN_1 | University of Leuven | 1 | 29 (14, 15) |
| LEUVEN_2 | University of Leuven | 1 | 35 (15, 20) |
| MAX_MUN | Ludwig Maximilians University Munich | 1 | 57 (24, 33) |
| NYU | NYU Langone Medical Center | 1 | 184 (79, 105) |
| OHSU | Oregon Healt and Science University | 1 | 28 (13, 15) |
| OLIN | Olin Neuropsychiatry Research Center | 1 | 36 (20, 16) |
| PITT | University of Pittsburgh School of Medicine | 1 | 57 (30, 27) |
| SBL | Social Brain Lab | 1 | 30 (15, 15) |
| SDSU | San Diego State University | 1 | 36 (14, 22) |
| STANFORD | Stanford University | 1 | 40 (20, 20) |
| TRINITY | Trinity Centre for Health Sciences | 1 | 49 (24, 25) |
| UCLA_1 | University of California Los Angeles | 1 | 114 (65, 49) |
| UCLA_2 | University of California Los Angeles | 1 | 27 (13, 14) |
| UM_1 | University of Michigan | 1 | 110 (55, 55) |
| UM_2 | University of Michigan | 1 | 35 (13, 22) |
| USM | University of Utah School of Medicine | 1 | 101 (58, 43) |
| YALE | Yale Child Study Center | 1 | 56 (28, 28) |
| BNI_1 | Barrow Neurological Institute | 2 | 58 (29, 29) |
| EMC_1 | Erasmus University Medical Center Rotterdam | 2 | 54 (27, 27) |
| ETH_1 | ETH Zurich | 2 | 37 (13, 24) |
| GU_1 | Georgetown University | 2 | 106 (51, 55) |
| IP_1 | Institute Pasteur | 2 | 56 (22, 34) |
| IU_1 | Indiana University | 2 | 40 (20, 20) |
| KKI_1 | Kennedy Krieger Institute | 2 | 211 (56, 155) |
| KUL_3 | Katholieke Universiteit Leuven | 2 | 28 (28, 0) |
| NYU_1 | NYU Langone Medical Center | 2 | 78 (48, 30) |
| NYU_2 | NYU Langone Medical Center | 2 | 27 (27, 0) |
| OHSU_1 | Oregon Healt and Science University | 2 | 93 (37, 56) |
| OILH_2 | Olin Institute of Living Hartford | 2 | 59 (24, 35) |
| SDSU_1 | San Diego State University | 2 | 58 (33, 25) |
| SU_2 | Stanford University | 2 | 42 (21, 16) |
| TCD_1 | Trinity Centre for Health Sciences | 2 | 42 (21, 21) |
| U_MIA_1 | University of Miami | 2 | 28 (13, 15) |
| **Total** | All sites | 1 and 2 | 2226 (1060, 1161) |

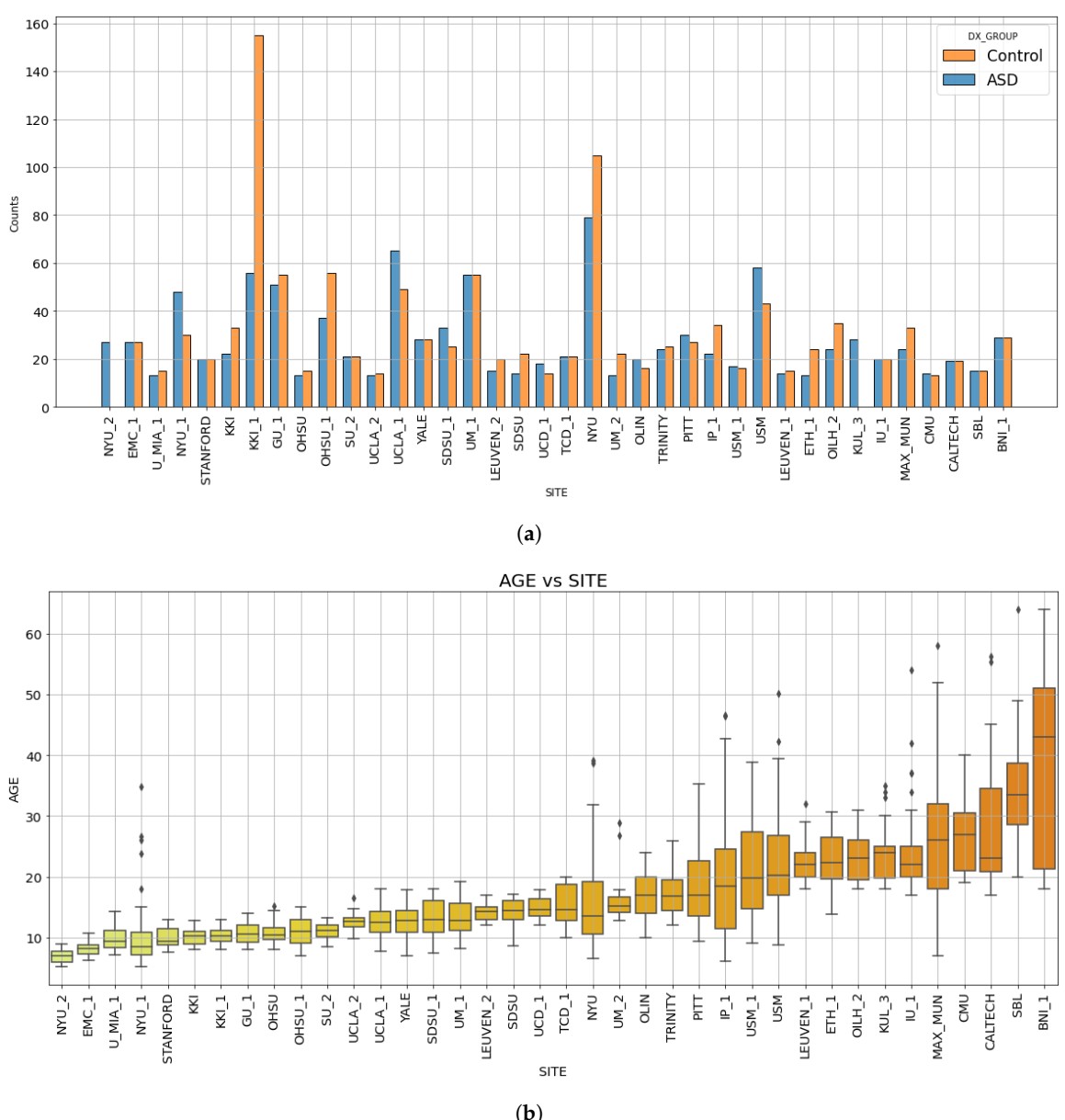

(**a**)

(**b**)

**Figure 1.** Bar diagram showing the number of subjects acquired at each site for each diagnostic group (**a**). Box plots showing the age distribution of subjects per site (**b**). In both plots, the sites are sorted by increasing median age. The subscripts 1 and 2 indicate the subsets acquired at that specific site released within either the ABIDE I or ABIDE II collections.

Because of the male vs. female unbalance in the ASD condition [16,17], which also occurs in the composition of the ABIDE datasets, we restricted our analysis to the cohort of male subjects. Moreover, due to the under-representation of subjects older than 40 years in the ABIDE datasets, only subjects with ages between 5 and 40 years were considered. We also excluded the sites NYU_2 and KUL_3 since, as noticeable from Figure 1a, they did not provide controls but only ASD subjects. In fact, without control data, it is not possible to run the harmonization procedure illustrated in Section 2.4. The list of the IDs of subjects of the ABIDE dataset used in this study is provided in the Supplementary Materials (Table S1). After adopting the selection criteria above, we obtained a dataset consisting of 1470 male subjects coming from ABIDE I and ABIDE II, of which 737 are TD controls and 733 ASD, from 36 sites.

### 2.2. Fmri Data Preprocessing

We processed the data using C-PAC, which is a configurable, open-source pipeline, based on the Nipype platform, which was proven to lead to better classification performances between ASD and TD subjects on the ABIDE I dataset [18]. The applied correction steps were the following: motion correction, slice timing correction, band-pass filtering, spatial smoothing and registration to a common template. At the end of functional preprocessing steps, the brain was parcellated using the Harvard-Oxford atlas (https://fsl.fmrib.ox.ac.uk/fsl/fslwiki/Atlases, accessed on 5 April 2023), obtaining 110 regions of interest (ROIs), from which average timeseries were extracted for each subject.

### 2.3. Functional Connectivity Measures

As a measure of connectivity, we used the Pearson correlation. It is the measure of a linear relation lying between two time series $x(t)$ and $y(t)$ and is defined as the ratio of the covariance of $(x, y)$ over the product of the standard deviation of the two sets. Given two discrete series $x = \{x_1, \ldots, x_n\}$ and $y = \{y_1, \ldots, y_n\}$ of length n, the correlation between them can be easily computed by

$$r_{xy} = \frac{\sum_{i=1}^{n} (x_i - \mu_x)(y_i - \mu_y)}{\sqrt{\sum_{i=1}^{n}(x_i - \mu_x)^2}\sqrt{\sum_{i=1}^{n}(y_i - \mu_y)^2}}. \tag{1}$$

The Pearson correlation coefficients were computed for each pair of timeseries of brain areas for all subjects of the ABIDE dataset. Each coefficient was then Fisher-z-transformed [19] in order to make its distribution quasi-normal:

$$z = \frac{1}{2}ln\left(\frac{1+r}{1-r}\right) = arctanh(r). \tag{2}$$

Following this procedure, it was possible to obtain $N_{comb} = n \cdot (n-1)/2 = 5995$ independent non-trivial combinations from $n = 110$ timeseries for each subject.

### 2.4. Data Harmonization with NeuroHarmonize and DNN Classification

The first classification analysis was implemented with a deep neural network analyzing the harmonized dataset. For the harmonization, we employed the state-of-art tool, developed by Pomponio et al. [20] in 2019 and available as a free Python package called NeuroHarmonize (https://github.com/rpomponio/neuroHarmonize, accessed on 5 April 2023). It differs from the previous formulation by Fortin [8] in the modeling of the biological covariates, which are usually treated with a linear regression. Pomponio substituted this linear model with a generalized additive model (GAM). In this model, covariates such as sex, age and full intelligence quotient (FIQ) are represented by terms that can be either linear or non-linear functions. This allows for better parametric modeling to deal with non-linear trends such as the typical evolution of several brain measures with age. The harmonized feature $y_{ijf}^*$ is given by

$$y_{ijf}^* = \frac{y_{ijf} - \hat{\alpha}_f - x_{ij} \cdot \hat{\beta}_f - \hat{\gamma}_{if}}{\hat{\delta}_{if}} + \hat{\alpha}_f + x_{ij} \cdot \hat{\beta}_f \tag{3}$$

where:

- $y_{ijf}$ is the numeric value of the feature $f$ for the patient $i$ acquired with the scan (or equivalently, from the site) $j$;
- $x_{ij}$ is the vector of the matrix X created with the covariates of interest, such as age and sex;
- $\hat{\alpha}_f$ is the estimator of the mean value for the feature $f$;
- $\hat{\beta}_f$ is the estimator of the vector of regression coefficients corresponding to $X$ for the feature $f$;

- $\hat{\gamma}_{if}$ and $\hat{\delta}_{if}$ represent the estimators of the additive and multiplicative terms for site-*i* effects related to feature *f*, respectively;
- $\epsilon_{ijf}$ is a residual term that is assumed to follow a normal distribution with zero mean and variance $\sigma_f^2$;

In our work, we chose the age and FIQ as covariates to be preserved in order to keep important possible biological trends in the data and avoid overcorrections.

In order to test the effectiveness of the harmonization procedure, we performed a classification task whose aim was to distinguish between different pair of sites by using the site name as label. We used a random forest classifier and the area under the receiver operating characteristic (ROC) curve (AUC) as a metric. AUC measures the overall performance of the model across all possible classification thresholds by calculating the area under the ROC curve. The AUC value ranges between 0 and 1, with a higher value indicating better classification performance. An AUC of 0.5 indicates that the model performs no better than random guessing, whereas an AUC of 1 indicates perfect classification performance [21]. We compared the performance of the classification for non-harmonized data and harmonized ones.

Harmonized data were then analyzed with a deep neural network (DNN), trained to perform the ASD vs. TD classification. The DNN was built using the Keras API and consisted of an input layer, a dense layer of 256 neurons, batch normalization, a dropout (with 30 % dropout probability), and 8-neuron dense layer. It ended with a single-neuron output layer, activated with an ReLU function. The results of ASD vs. TD discrimination were evaluated in terms of the mean AUC scores across all the folds in a 5-fold CV scheme.

### 2.5. Adversarial Neural Network for Embedded Data Harmonization and Classification

We proposed a different approach to the analytical harmonization by developing a DNN model specifically designed to perform a classification unbiased by site-related features.

The construction of this network, which is schematically represented in Figure 2 and referred to as ad-DNN, obtains its main idea from the network proposed by Ganin et al. [11]. We replicated the structure of their model, which consists of three parts. A feature extractor branch takes input data and creates an inner representation of them within its structure; we modeled this part as dependent on parameters $\theta_f$, which represent the weights of all the connections between neurons. The network is then forked into two branches: a label classifier branch for control vs. ASD classification, dependent on parameters $\theta_y$ and with an associated loss function $L_y$, and a site classifier branch aimed for site classification, which is dependent on parameters $\theta_d$ and is associated to a loss function $L_d$. The optimization of the overall loss function $E(\theta_f, \theta_y, \theta_d)$ will then depend on all the parameters constituting the network because it can be written as:

$$E(\theta_f, \theta_y, \theta_d) = L_y(\theta_f, \theta_y) - \lambda L_d(\theta_f, \theta_d).$$ (4)

In the training phase of this particular network, parameters are updated in order to obtain a minimization of the loss function $E(\theta_f, \theta_y, \theta_d)$ with respect to parameters $\theta_f$ and $\theta_y$, and a maximization of it with respect to $\theta_d$; in other words, one seeks for a saddle point given by:

$$\hat{\theta}_f, \hat{\theta}_y = \underset{\theta_f, \theta_y}{argmin} E(\theta_f, \theta_y, \theta_d)$$

$$\hat{\theta}_d = \underset{\theta_d}{argmax} E(\theta_f, \theta_y, \theta_d).$$ (5)

If compared to the approach described in [11], our model presents a substantial difference in the structure of the domain-classifier branch. Unlike their work, which only concerns 2 domains, our model has to deal with 36 sites because, as described in Section 2.1, data from the ABIDE datasets are collected from 36 medical centers.

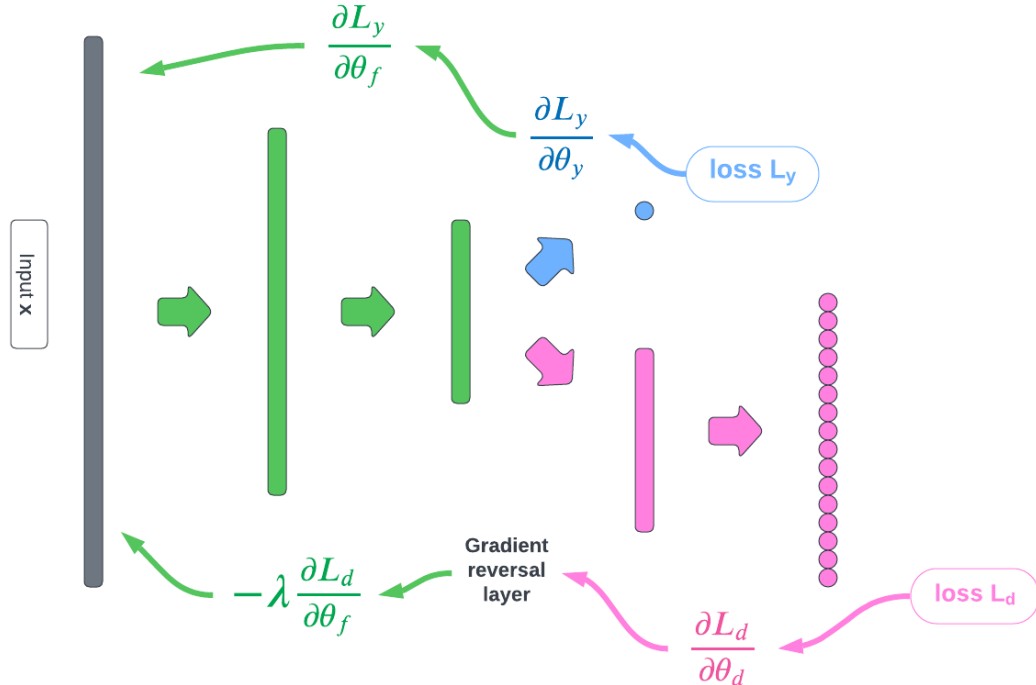

**Figure 2.** Schematic representation of the ad-DNN adversarial network employed in this work for ASD vs. TD classification, and the main gradients that play a relevant role in the weight update. The neural network layers are pictured as bars whose length is representative of the number of neurons in each layer. The green bars constitute the feature extractor branch, which forks into the label classifier branch (a single-output neuron) in blue, and the site classifier in pink, consisting of a hidden layer and a multiple-output layer. The main gradients are highlighted with a color matching the section of the network that they operate on.

The structure of the ad-DNN adversarial network, which is depicted in Figure 3, is similar to the one of DNN described in the previous section but, whereas the DNN ends with a single-neuron output layer, the adversarial network is split at this point into the label classifier layer consisting of a single neuron, and the site classifier branch. At the beginning of the site classifier, the gradient reversal layer is placed, linked to a dense layer with 16 neurons, batch normalization and the final output dense layer consisting of 36 neurons as the number of sites. To restrain overfitting, this configuration was obtained after different attempts to create a model that gives the maximum reachable performances with the minimum number of parameters. These results, as in the previous analysis, are reported as the mean AUC scores across all the folds in a 5-fold CV scheme.

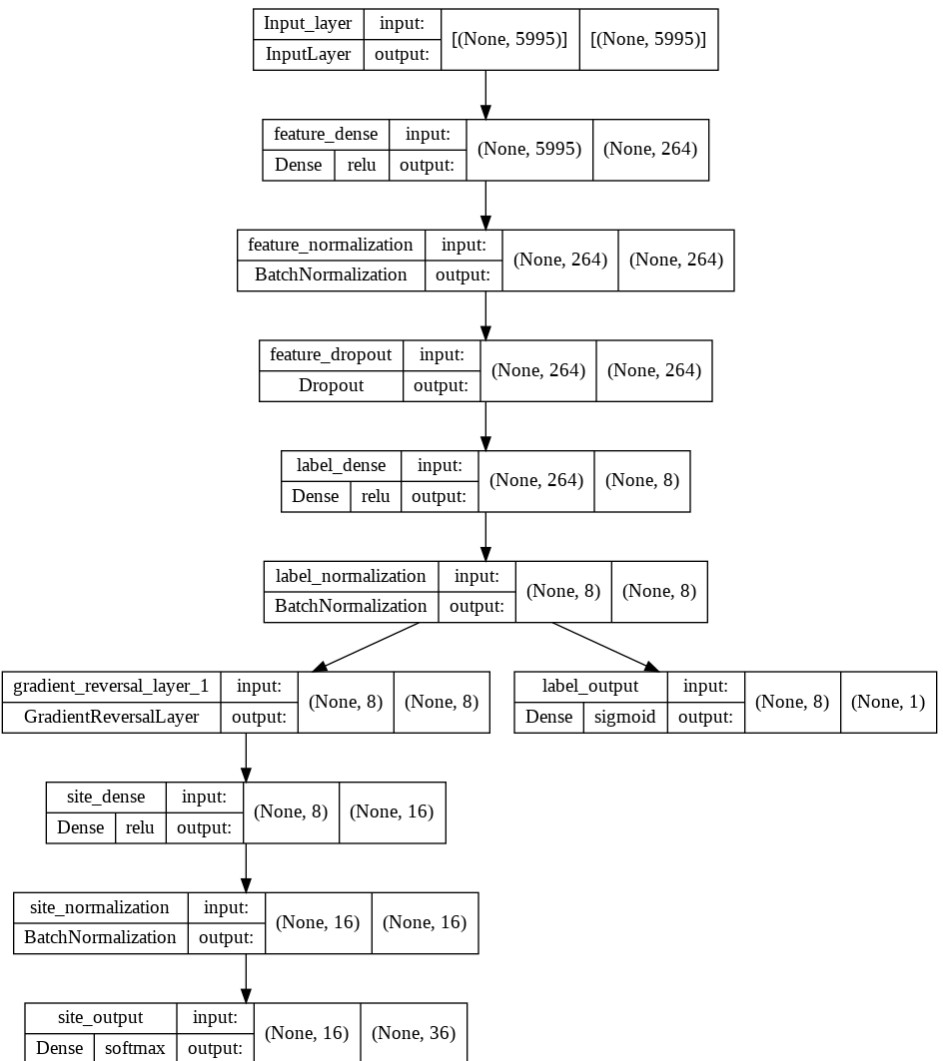

**Figure 3.** Architecture of the domain-adversarial neural network (ad-DNN) with two outputs: a single-neuron output for binary classification of ASD vs. TD and a multi-class classification for site classification. Each box contains the layer name and the related task (Input, Dense, BatchNormalization, etc.), activation function, input shape (number of neurons) and output shape.

### 2.6. Identification of Relevant Brain Areas with SHAP

SHAP (https://github.com/slundberg/shap, accessed on 5 April 2023), which stands for SHapley Additive exPlanation, is a technique based on game theory that provides a method for machine learning model explainability. The idea behind SHAP is to use Shapley values [22,23] to express the impact of each input data feature to a model output.

DeepSHAP (https://shap-lrjball.readthedocs.io/en/latest/generated/shap.DeepExplainer.html, accessed on 5 April 2023) [24] is a model-specific explanatory algorithm specifically designed to perform on deep neural networks.

Since Shapley values can be positive or negative, the absolute importance value for a feature $f$ is calculated as the mean of the magnitude of all the Shapley values for that feature, so the mean is performed across all the $n$ samples under examination. Thus, the absolute importance value can be expressed as:

$$I_f = \frac{1}{n} \sum_{i=1}^{n} \|\phi_i(f)\| . \tag{6}$$

Each feature is representative of a correlation between two brain areas. In this paper, we investigated what the most recurrent brain areas are that have a role in the discrimination between subjects with ASD and TD controls.

## 3. Results

### 3.1. Data Harmonization with NeuroHarmonize

Regarding the evaluation of the efficacy of the NeuroHarmonize protocol for multi-site data harmonization, we report in Figures 4 and 5 the classification performances obtained with a random forest classifier in site vs. the site binary classification of connectivity measures.

It is possible to notice that, before the harmonization of data (Figure 4), the AUC values are mainly close to 1, which stands for the exact ability to distinguish between two sites. The AUC values decrease up to the expected ∼0.5 value when the classification is performed using harmonized data (Figure 5). This stands as a confirmation that the harmonization procedure removes (or at least reduces) site-related effects on features, making subjects acquired at different sites less recognizable.

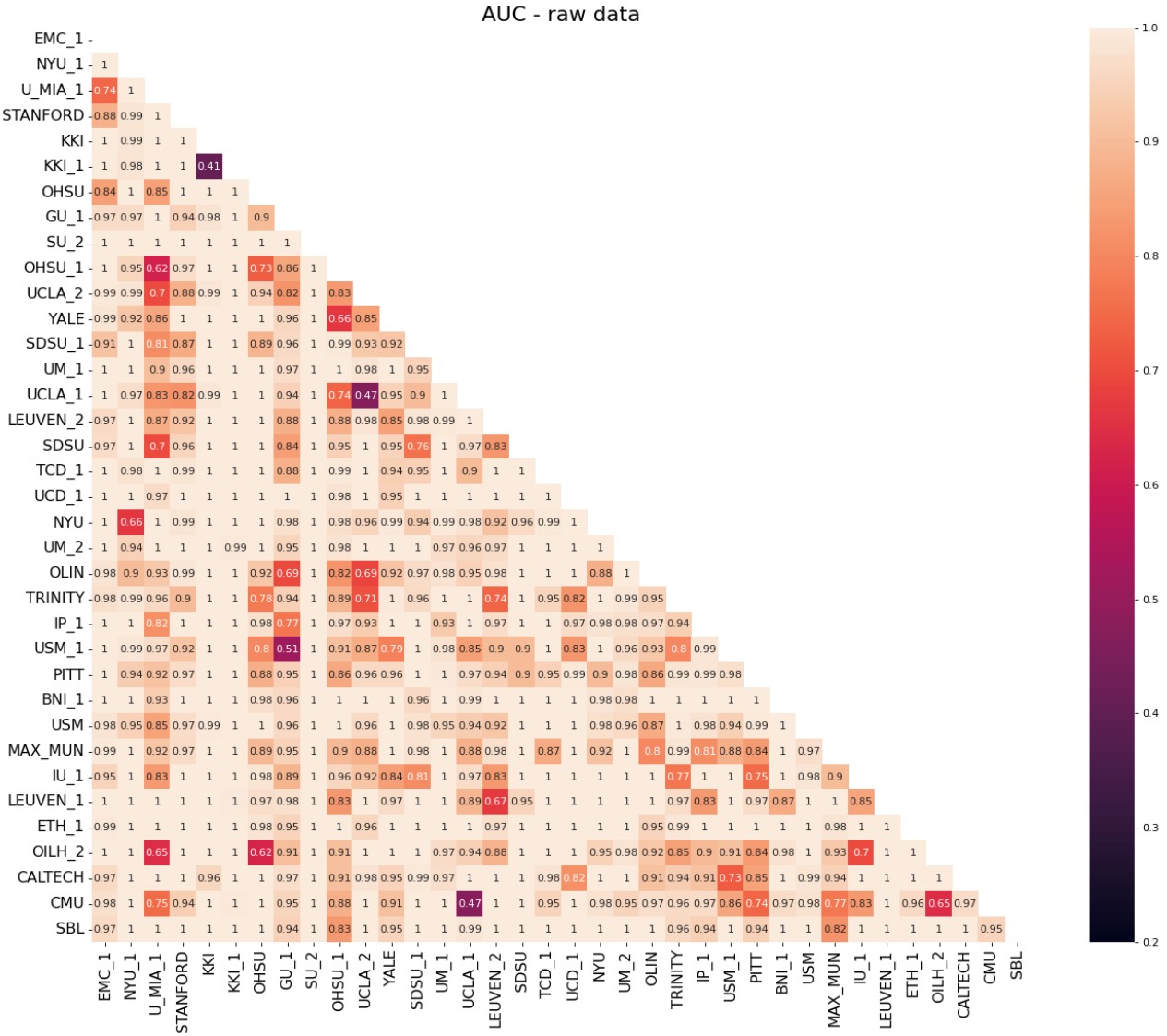

**Figure 4.** Heatmap of the AUC scores obtained in the binary classification of site vs. site for non-harmonized data.

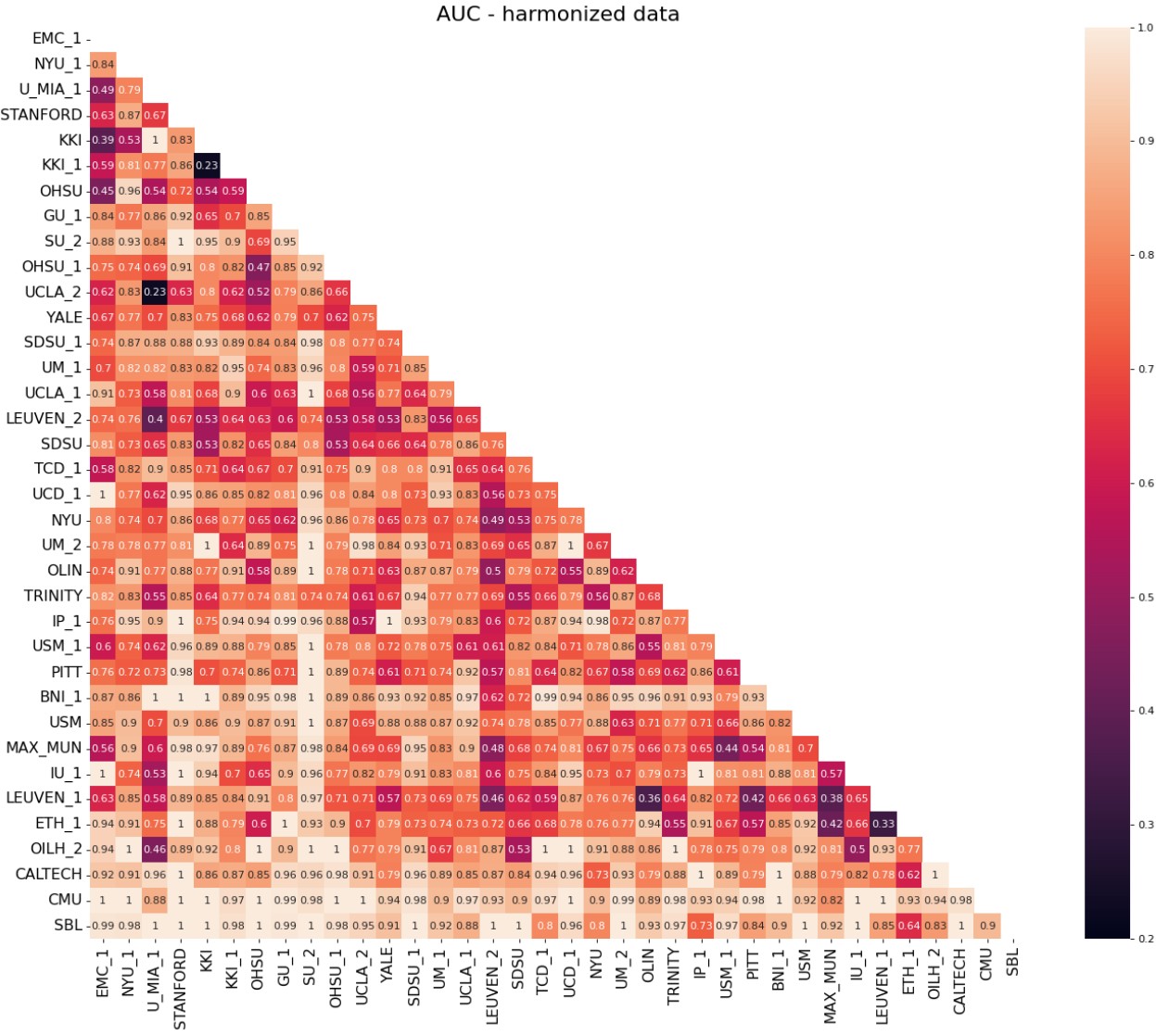

**Figure 5.** Heatmap of the AUC scores obtained in the binary classification of site vs. site for harmonized data.

### 3.2. Classification Results

The classification of harmonized data using the deep neural network described in Section 2.4 gave an AUC = $0.71 \pm 0.01$. The classification of non-harmonized data with the ad-DNN yields similar results: AUC = $0.70 \pm 0.03$. These results are in line with other results reported in the literature on the same topic.

### 3.3. Consistency between Relevant Features Identified by the Two Different Approaches

Using DeepSHAP, we are able to analyze the networks after their training in order to understand if there are some features with more influence than others during the prediction of the data label.

In Figure 6, we show two plots listing the first 20 most important features, sorted by importance, used by the neural network for the classification of harmonized data (Figure 6a), and the features extracted from the adversarial network with the analysis of raw data (Figure 6b). Through this graphic representation, it is easy to notice that some features are common between the two approaches. This implies that some altered functional connections have a greater impact than others on the characterization of ASD.

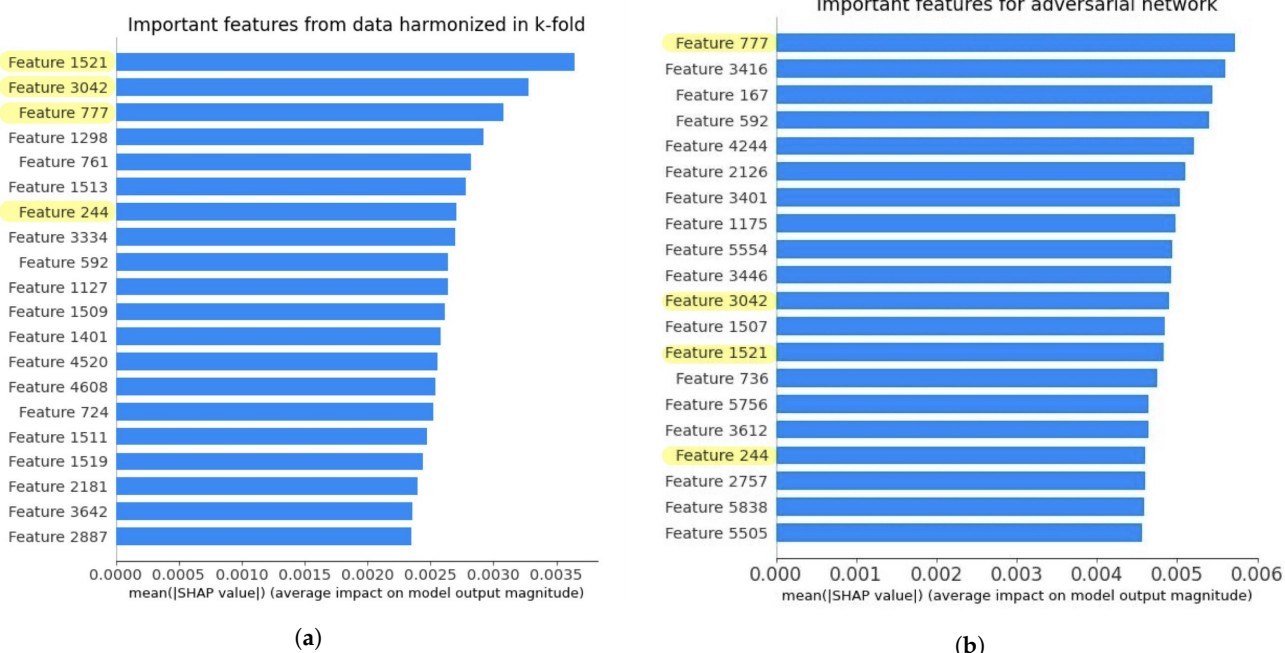

(**a**)                    (**b**)

**Figure 6.** Bar plots of the first 20 most important features in the ASD discrimination, according to the SHAP method applied to data harmonized and classified with: (**a**) the NeuroHarmonize followed by a DNN; (**b**) the end-to-end adversarial learning approach provided by the ad-DNN. Common items in the two sets of relevant features are highlighted in yellow.

To define the biological brain areas involved, we studied the occurrences of the ones involved in the most relevant altered connections. The result can be visualized in the histogram displayed in Figure 7 showing the occurrences of each area for the first 60 features extracted from the classification of harmonized data with the DNN and of raw data with the adversarial network pooled together.

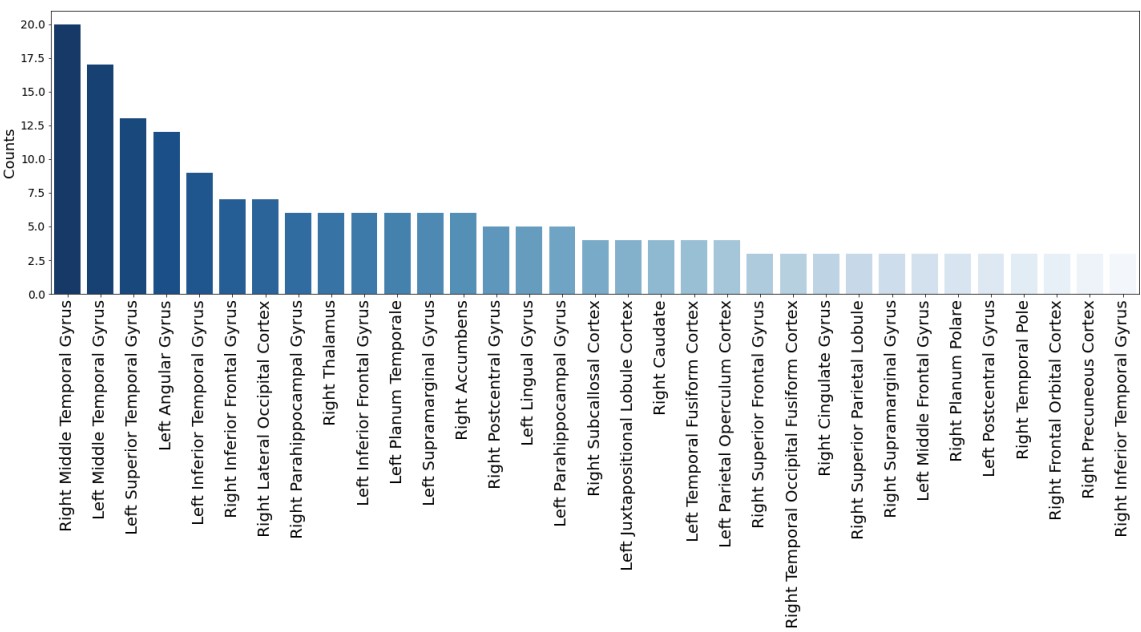

**Figure 7.** Histogram of the most important brain areas obtained by merging the results of the important features extracted using SHAP on the two neural networks employed in the analyses.



The list of the encoded name for each functional correlation coefficient and the corresponding pair of brain areas generating it is provided in the Supplementary Materials (Table S2).

## 4. Conclusions and Discussion

The goal of the present study was to apply an adversarial neural network to the case study of ASD and perform classification using multi-site functional connectivity data. The second objective was to compare this approach of multi-site data harmonization to the analytical and more commonly used one based on the NeuroHarmonize tool.

This latter approach was carried out in the study by Ingalhakinar et al. [25]. In their paper, working on the ABIDE I dataset, and parcellating the brain with a different atlas called CC200, they obtained, after data harmonization, AUC values of 0.66 and 0.8 depending on the classifier employed for the analysis.

In the recent paper by Kang et al. [26], the authors proposed a different approach to the analysis of timeseries and subsequent classification of data from the ABIDE I dataset. The authors used a long short-term memory (LSTM) network to extract correlations and features from cerebral ROIs, and then they classified these features with a deep neural network after applying data augmentation techniques. They also compared their approach to the standard analytical method based on Pearson's correlation coefficients between the timeseries and dimensionality reduction with PCA. Their approach differs from ours in the use of an LSTM network and PCA. Furthermore, unlike our paper, their study did not address the multi-site harmonization problem. Remarkably, despite their quite different approach from ours, Kang et al. achieved similar results to ours, obtaining an AUC of 72% on non-harmonized data in a cross-validation scheme.

In our study, the two different approaches to multi-site data harmonization illustrated above lead us to consistent results on the whole ABIDE (I and II) dataset, namely AUC = $0.71 \pm 0.01$ for the DNN analysis of harmonized data and AUC = $0.70 \pm 0.03$ with the adversarial network. We are prone to consider the second one as a more promising strategy since it is completely data-driven and does not need any data preparation before inputting them to a machine learning algorithm.

However, a possible drawback of using this type of network with data such as these is the possible confusion in the weights updating introduced by the huge number of sites and the limited dimension of the dataset from each site, which does not allow for an accurate training of the network.

In different papers, domain-adversarial neural networks outperform different domain adaptation techniques; however, so far, these networks have only been tested on two domains [11–13]. In the article, Ganin et al. [11], who first designed this network, tested their algorithm on different datasets, which include two different domains, each one containing labeled or unlabeled data. In this way, they were able to test the domain adaptability of their network and, depending on the analyzed dataset, they achieved improvements in accuracy of up to 20%. In the study by Kamath et al. [12], a DANN was used for the analysis of text datasets. This approach outperforms other "domain adaptation" techniques, leading to an improvement of 2–7% accuracy. In the work of Guan et al. [13], an improvement in accuracy using a DANN of more than 8% compared to other techniques was reported. In all these papers, the so-called *domains* are the equivalent of our *sites*.

However, in our data, we dealt with 36 different sites. Even if the site-predictor branch is encouraged to learn site-distinctive traits, it is not always able to distinguish among so many different sites. This is mainly due to the small size of some sites, and can cause confusion during the back-propagation on the update of weights related to the feature extractor branch. Also taking into account the previously described limitations, the ad-DNN approach obtained comparable performances with the analytical harmonization of connectivity measures followed by a DNN. The advantage of the former approach is that it is completely data-driven, with no need for preliminary data processing.

Certainly, the proposed approach of classification with an ad-DNN could represent a promising tool to employ in a dataset of great dimensions. A crucial issue of this work is in fact the small amount of data acquired at each site, which compose the large dataset.

To determine the consistency of the brain areas found in our analysis with some similar studies, we compared the results with the ones obtained by Saponaro et al. [14] from a structural analysis of the ABIDE dataset. The areas of temporal gyri are among the most discriminant regions: they belong to the temporal lobe, which is involved in processes such as language comprehension, emotion recognition and social cognition, and whose dysfunction has been consistently reported in ASD [27,28]. Left angular and left supermarginal gyri are part of the inferior parietal lobule, a region implicated in impaired social-communicative and motor functions [29], including the processing of pragmatic information [30] and sentence comprehension [31] in individuals with ASD. Regions such as the right orbitofrontal and right accumbens are both involved in reward and emotion. Previous investigations detected an impaired function of these reward pathways when individuals with ASD viewed smiling faces [32] and heard human voices [33], supporting the impaired salience and reward value attributed to social stimuli in autism [34]. The left lingual gyrus, together with other brain regions—lateral occipital cortex, fusiform gyrus and posterior superior temporal sulcus—is part of a complex network implicated in atypical object/face recognition and processing biological motion in ASD [35].

As highlighted in this latter analysis, both the procedures, i.e., the deep learning classification of analytically harmonized data and the deep adversarial learning, lead to a common set of important features. This similarity became even more evident when we converted functional connectivity measures into the brain areas mainly involved.

We observe that a great number of areas that we found as relevant in our discrimination problem based on brain functional data were actually identified as also important in a recent study [14] carried out on the structural MRI data, and that they are related to ASD symptoms. It is important to note that, while the AUC performance achieved indicates an acceptable discrimination capability that can reveal information on the neural correlates of ASD, the method presented in this work is not suitable for diagnostic purposes. In order to improve the prediction accuracy, other approaches should be considered, including the combination of genetic and imaging information as well as the application of approaches based of the deviation from the normative model, as already performed in other psychiatric disorders [36].

As a final consideration, we highlight that, despite the case study reported in this work being focused on the investigation of the ASD condition through fMRI data, the data analysis pipeline developed here constitutes a general procedure that can be extended to other disease and other patients' data acquisition modalities. In particular, the adversarial deep neural network approach presented here can not only be easily extended to the investigation of different conditions where an altered functional connectivity is suspected, but can be exploited in all classification or regression problems where confounding effects such as those related to acquisition instrumentation (including other MRI modalities, X-ray-based techniques, EEG, MEG, etc.) can affect the results.

**Supplementary Materials:** The following supporting information can be downloaded at: https://www.mdpi.com/article/10.3390/app13116486/s1, Table S1: containing the list of subjects of the ABIDE dataset analyzed; Table S2: containing the look-up table for converting the connectivity feature numbers into the involved pairs of brain regions.

**Author Contributions:** Conceptualization, A.R.; methodology, A.R., S.C. and P.O.; software, F.C.; validation, P.O.; formal analysis, F.C.; investigation, F.C.; resources, A.R. and P.O.; data curation, P.O.; writing—original draft preparation, F.C.; writing—review and editing, A.R., S.C. and P.O.; supervision, A.R., S.C. and P.O.; project administration, A.R.; funding acquisition, A.R. and P.O. All authors have read and agreed to the published version of the manuscript.

**Funding:** This research was funded by: the National Institute for Nuclear Physics (INFN), next_AIM (Artificial Intelligence in Medicine: next steps) research project (INFN-CSN5), https://www.pi.infn.it/aim (accessed on 5 April 2023); Italian Ministry of Health, *Ricerca Corrente* and 5 × 1000 voluntary contributions to IRCCS Fondazione Stella Maris; EU, AIMS-2-Trials, https://www.aims-2-trials.eu (accessed on 5 April 2023).

**Data Availability Statement:** The data used in this work have been collected within the ABIDE research project and made publicly available at https://fcon_1000.projects.nitrc.org/indi/abide, accessed on 5 April 2023.

**Conflicts of Interest:** The authors declare no conflict of interest.

## Abbreviations

The following abbreviations are used in this manuscript:

| | |
|---|---|
| AI | Artificial Intelligence |
| ASD | Autism Spectrum Disorders |
| AUC | Area Under the ROC Curve |
| ad-DNN | Site-Adversarial Deep Neural Network |
| DNN | Deep Neural Network |
| FIQ | Full Intelligence Quotient |
| GAM | Generalized Additive Model |
| ML | Machine Learning |
| rs-fMRI | resting-state functional Magnetic Resonance Imaging |
| TD | Typically Developing control subjects |

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
