# Peer review of "Multi-Site MRI Data Harmonization with an Adversarial Learning Approach: Implementation to the Study of Brain Connectivity in Autism Spectrum Disorders"

_applsci, doi:10.3390/app13116486_

Round 1
Reviewer 1 Report
The submitted manuscript is nicely written, concise, interesting, presents sufficient amount of new information. In my opinion, it can be accepted if the Authors include the solutions to the questions raised below in the revised version.
Line 16, AUC should be explained here
Lines 21-22, keywords must be provided
Lines 168-170, here some information on AUC should be provided, i.e. what the AUC values informs us about
Figure 2, it should be explained what are the meanings of the colors applied as well as the shape (i.e. circles vs bars)
Figure 3 should not be placed in the middle of the sentence
Figure 6, if the same feature is present in both (a) and (b) it should have the same color (not blue). It would be informative.
Lines 317-318, a reference to this mentioned study should be provided here
At the end of the manuscript the Authors should include the section describing each Author’s individual contribution. This is mandatory in Applied Sciences journal.
Minor editing of English language required
Author Response
Please find below the point-by-point responses to your comments. The corresponding changes introduced in the manuscript are highlighted as red text in the revised version.
The submitted manuscript is nicely written, concise, interesting, presents sufficient amount of new information. In my opinion, it can be accepted if the Authors include the solutions to the questions raised below in the revised version.
Line 16, AUC should be explained here
We added a short definition of AUC in the abstract, whereas a more detailed description is added in the method section (see the response to the pertaining point below).
Lines 21-22, keywords must be provided
We thank the reviewer for highlighting this point. We added the following keywords to the manuscript: "Brain connectivity; machine learning; adversarial learning; Autism Spectrum Disorders (ASD); multi-site harmonization; explainable AI (XAI)".
Lines 168-170, here some information on AUC should be provided, i.e. what the AUC values informs us about
We added where indicated by the reviewer the definition of the AUC and the meaning of the values it can assume. We added also a reference to the article by Fawcett, T. (2006) [An introduction to ROC analysis. Pattern recognition letters, 27(8), 861-874. doi: 10.1016/j.patrec.2005.10.010].
Figure 2, it should be explained what are the meanings of the colors applied as well as the shape (i.e. circles vs bars)
We improved the caption of the figure to clarify the meaning of the colors and of the symbols. We modified the text as follows:
"The neural network layers are pictured as bars whose length is representative of the number of neurons in each layer.
The green bars constitute the feature extractor branch which forks into the label classifier branch (a single-output neuron) in blue, and the site classifier in pink, consisting of a hidden layer and a multiple-output layer. The main gradients are highlighted with a color matching the section of the network they operate on".
Figure 3 should not be placed in the middle of the sentence
We changed the position of Figure 3 within the text.
Figure 6, if the same feature is present in both (a) and (b) it should have the same color (not blue). It would be informative.
We appreciated this comment by the reviewer, which allows us to present the information more clearly. We highlighted in the figure the features present in both (a) and (b) and we specified this choice in the figure caption.
Lines 317-318, a reference to this mentioned study should be provided here
We thank the reviewer for pointing this out. We wanted to indicate the reference to the article by Saponaro et al. (2022) [A Multi-site harmonization of MRI data uncovers machine-learning discrimination capability in barely separable populations: An example from the ABIDE dataset. NeuroImage: Clinical 2022, 35, 103082. https://doi.org/10.1016/j.nicl.2022.103082], recently published by our group, but we forgot to insert it during the drafting of the manuscript. We added it to the manuscript.
At the end of the manuscript the Authors should include the section describing each Author’s individual contribution. This is mandatory in Applied Sciences journal.
We thank the reviewer for highlighting this point. We added the following paragraph to the manuscript:
"Author Contributions: Conceptualization, A.R.; methodology, A.R, S.C. and P.O.; software, F.C.; validation, P.O.; formal analysis, F.C.; investigation, F.C.; resources, A.R. and P.O.; data curation, P.O.; writing—original draft preparation, F.C.; writing—review and editing, A.R, S.C. and P.O.; supervision, A.R, S.C. and P.O.; project administration, A.R.; funding acquisition, A.R. and P.O. All authors have read and agreed to the published version of the manuscript."
Reviewer 2 Report
Authors present a study on use of adversial deep learning approach to study brain connectivity in multi-site MRI data harmonization in patients with autism spectrum disorder by comparing two different approaches: the analytical ComBat-GAM procedure, and site-adversarial deep neural network (ad-DNN), which performs classification and searches for site-relevant patterns, to make predictions free from site-related biases. ad-DNN achieved the performance comparable with that obtained by a DNN on data previously harmonized; the connectivity alterations identified in both procedures showedan agreement between each other and with the patterns of neuroanatomical alterations previously detected.
Introduction is well written and provides enough information of the subject. Keywords are missing (labeled as keyword 1,2,3...). For Materials and Methods, a physicist is required to asses the methods of data harmonization , classification as well as identification of brain areas with AI SHAP. Results are written in understandable manner. There is virtually not a single section or paragraph on clinical utility of this experiment, i.e possible use for the diagnostic and treatment of these patients as well as for future advances.
There is a recent similar study on the subject:
Kang L, Chen J, Huang J, Jiang J. Autism spectrum disorder recognition based on multi-view ensemble learning with multi-site fMRI. Cogn Neurodyn. 2023 Apr;17(2):345-355. doi: 10.1007/s11571-022-09828-9. Epub 2022 Jun 17. PMID: 37007200; PMCID: PMC10050260.
I suggest to compare the results with this research and extensive comment.
Acceptable.
Author Response
Please find below the point-by-point responses to your comments. The corresponding changes introduced in the manuscript are highlighted as red text in the revised version.
Introduction is well written and provides enough information of the subject. Keywords are missing (labeled as keyword 1,2,3...).
We added the following keywords to the manuscript: Brain connectivity; machine learning; adversarial learning; Autism Spectrum Disorders (ASD); multi-site harmonization; explainable AI (XAI).
For Materials and Methods, a physicist is required to asses the methods of data harmonization , classification as well as identification of brain areas with AI SHAP. Results are written in understandable manner. There is virtually not a single section or paragraph on clinical utility of this experiment, i.e possible use for the diagnostic and treatment of these patients as well as for future advances.
We thank the reviewer for these comments that provided us the opportunity to clarify these points. The relevance of the results achieved by the analysis pipeline we implemented are evaluated and discussed by the author S.C., who is a neuropsychiatrist with specific expertise in brain imaging. To better highlight the limited clinical utility of this work we added the following considerations in the discussion paragraph:
"It is important to note that, while the AUC performance achieved indicate an acceptable discrimination capability that can reveal information on the neural correlates of ASD, the method presented in this work is not suitable for diagnostic purposes. In order to improve the prediction accuracy, other approaches should be considered, including the combination of genetic and imaging information as well as the application of approaches based of the deviation from normative model, as already performed in other psychiatric disorders [Oliveira-Saraiva, D.; Ferreira, H.A. Normative model detects abnormal functional connectivity in psychiatric disorders. Frontiers in Psychiatry 2023, 14. https://doi.org/10.3389/fpsyt.2023.1068397.]."
There is a recent similar study on the subject:
Kang L, Chen J, Huang J, Jiang J. Autism spectrum disorder recognition based on multi-view ensemble learning with multi-site fMRI. Cogn Neurodyn. 2023 Apr;17(2):345-355. doi: 10.1007/s11571-022-09828-9. Epub 2022 Jun 17. PMID: 37007200; PMCID: PMC10050260.
I suggest to compare the results with this research and extensive comment.
We thank the reviewer for having pointed out the interesting paper by Kang et at (2023). We added the following considerations in the discussion paragraph of the paper:
"In the recent paper by Kang \textit{et al.}~\cite{kang2023}, the authors proposed a different approach to the analysis of timeseries and subsequent classification of data from the ABIDE I dataset. The authors used a Long Short-Term Memory (LSTM) network to extract correlations and features from cerebral ROIs, and then they classified these features with a deep neural network, after applying data augmentation techniques. They also compared their approach to the standard analytical method based on Pearson's correlation coefficients between timeseries and dimensionality reduction with PCA. Their approach differs from ours in the use of a LSTM network and PCA. Furthermore, unlike our paper, their study did not address the multi-site harmonization problem. Remarkably, despite their quite different approach from ours, Kang et al. achieved similar results to ours, obtaining an AUC of 72% on non-harmonized data in a cross-validation scheme."
Reviewer 3 Report
This paper presents a solution based on adversarial deep neural network (ad-DNN) for multicenter MRI data harmonization, in comparison to a previously reported and validated analytical ComBat-GAM procedure.
It is a good article, well written and easy to follow. The paper is very well structured into sub-sections. This improves readability.
The issue targeted in the paper, namely the suppression of biases, is of utmost importance, and can be extrapolated to publicly available databases of ECG,EMG, gait pattern, etc. Although it is not mandatory, and in the end it is the decision of the authors, I would appreciate a discussion in this direction.
I have some minor remarks:
1) the introduction only presents this study shortly, lines 86-92. Please provide more detail about the proposed work
2) section 2.1, please present the dataset in a tabular form
3) secton 2.5, it is clear here that the authors porpose teh solution based on adversial networks. this should also be made clear in the introduction
The methodology is very well and clearly described, teh results are credible and support the claims.
I have only noticed some slight language errors which can be corrected during a final proofreading before revision
Author Response
Please find below the point-by-point responses to your comments. The corresponding changes introduced in the manuscript are highlighted as red text in the revised version.
The issue targeted in the paper, namely the suppression of biases, is of utmost importance, and can be extrapolated to publicly available databases of ECG,EMG, gait pattern, etc. Although it is not mandatory, and in the end it is the decision of the authors, I would appreciate a discussion in this direction.
We thank the reviewer for these appreciations of the work. To emphasize the concept of the portability of this approach to analysis of different datasets affected by similar confounding effects related to data acquisition, we rephrased and extended the final sentence of the manuscript as follows:
"As a final consideration, we highlight that, despite the case study reported in this
work is focused of the investigation of the ASD condition through fMRI data, the data analysis pipeline developed here
constitutes a general procedure, which can be extended to other disease and other patients' data acquisition modalities. In particular, the adversarial deep neural network approach presented here can not only be easily extended to the investigation of different conditions
where an altered functional connectivity is suspected, but it can be exploited in all classification or regression problems where confounding effects such as those related to the acquisition instrumentation (including other MRI modalities, X-ray based techniques, EEG, MEG, etc.) can affect the results."
I have some minor remarks:
1) the introduction only presents this study shortly, lines 86-92. Please provide more detail about the proposed work
In this study, we compare two different approaches to the classification of functional connectivity data of subjects with ASD and TD, while mitigating biases related to the acquisition instrumentation (batch effect).
As described in section 2.3, the first step involves the computation of a brain connectivity measure known as Pearson correlation. Starting from these measures, we follow two different techniques.
The first approach, as detailed in section 2.4, involves the analytical harmonization technique known as ComBat which we implemented by means of its state-of-the-art improved version, NeuroHarmonize. The approach we propose utilizes statistical corrections to remove site-related biases and involves the development of a deep neural network for the classification of harmonized data. The second approach, as detailed in section 2.5, involves the development and use of a domain-adversarial neural network. This approach aims to discriminate between ASD and TD features while minimizing the biases caused by the sites through a learning process that minimizes the information learned from each site.
In addition, we implemented an explainable AI algorithm called SHAP, as described in section 2.6, to identify the brain areas that have more influence on the outcome of the deep learning classifier and quantify the importance of each feature. By identifying the altered functional connections related to these areas, we can determine the distinctive traits of subjects with ASD.
Overall, our study aims to provide an insight into the performance and effectiveness of the two different approaches in mitigating site-related biases and classifying subjects with ASD with respect to control subjects.
2) section 2.1, please present the dataset in a tabular form
We added a table (Table 1) to report schematically the composition of the dataset, which is constituted by the ABIDE I and ABIDE II cohorts, each one collecting data from different sites. For each site, the number or controls and ASD subjects are provided.
3) section 2.5, it is clear here that the authors purpose thu solution based on adversarial networks. this should also be made clear in the introduction
We agree with the reviewer and we clarified and extended the description of the purpose of the paper in the introduction, as specified in the response to point 1).
Round 2
Reviewer 2 Report
Authors have sufficiently responded to remarks.
Acceptable.